



# Brief communication: the potential use of low-cost acoustic sensors in short-term urban flood warnings

Nadav Peleg[1,*], Herminia Torelló-Sentelles[1,*], Grégoire Mariéthoz[1], Lionel Benoit[2], João P. Leitão[3], and Francesco Marra[4]

[1]Institute of Earth Surface Dynamics, University of Lausanne, Lausanne, Switzerland
[2]Biostatistics and Spatial Processes (BioSP), INRAE, Avignon, France
[3]Department of Urban Water Management, Swiss Federal Institute of Aquatic Science and Technology, Dubendorf, Switzerland
[4]Institute of Atmospheric Sciences and Climate, National Research Council of Italy (CNR-ISAC), Bologna, Italy
[*]These authors contributed equally to this work

**Correspondence:** Nadav Peleg (nadav.peleg@unil.ch), Herminia Torelló-Sentelles (herminia.torello@unil.ch)

**Abstract.** Floods in urban areas are one of the most common natural hazards. Due to climate change enhancing extreme rainfall, and cities becoming larger and denser, the frequency, magnitude and impact of these events are expected to increase. Pluvial floods can occur in urban areas within minutes. A fast and reliable flood warning system should thus be implemented in flood-prone cities to warn the public of upcoming floods and save lives and reduce damage. The purpose of this brief communication is to discuss the potential implementation of low-cost acoustic rainfall sensors in short-term flood warning systems.

## 1 Introduction

Most of the damage costs caused by natural hazards in cities are related to floods, one of the most common natural hazards (Kreibich et al., 2014). There are two types of urban flooding caused by rainfall: fluvial and pluvial. A fluvial flood occurs when a river flows over its banks or when a lake level rises over its shores. In most cases, it is caused by a prolonged event with medium- to high-intensity rainfall, which is not necessarily extreme, and therefore has a medium to high forecast potential (Richardson et al., 2020). A pluvial flood occurs when a large volume of rain falls during a short period of time, and exceeds the capacity of the urban drainage system. There is greater difficulty forecasting short-duration and intense rainfalls that trigger pluvial floods, because they are often convective in nature and can initiate directly over the urban area (Rözer et al., 2021).

There has been an increase in urban pluvial flood damage in recent years, and this trend is likely to continue in the future (Paprotny et al., 2018). The main reason for this is twofold. First, cities are becoming larger and denser, which implies that the extent of impervious surface grows bigger, increasing fast runoff generation, and that there are more population and infrastructure that may be vulnerable to flooding. Second, global warming is increasing the frequency and intensity of short-duration heavy rainfall events (Westra et al., 2014; Ali et al., 2021). In addition, the presence of high aerosol concentrations and extensive heat flux in cities can also enhance the intensification of heavy rainfall even further (Huang et al., 2022).



Flood early warning systems help mitigating flood impacts by reducing population and authorities unpreparedness, and therefore have the potential to save lives and reduce damages. Short-term forecast systems typically combine numerical models (weather forecasting, precipitation nowcasting, and hydrodynamic) with ground and remote sensing observations of precipitation (weather radars and satellites), and are mainly operated by national meteorological or hydrological agencies. For example, the Bureau of Meteorology in Australia provides short-term rainfall forecasts and alerts at the continental scale (Bowler et al., 2006), while MeteoSwiss provides short-term extreme rainfall alerts at the country level (Sideris et al., 2020). A functional early warning system should not only forecast the upcoming flood, but also communicate it rapidly and easily to the public using the media (e.g., radio and television) and mobile phones (using specially designed applications or SMS push messages).

The World Meteorological Organization estimates that one-third of the world's population is still unprotected by early warning systems, mainly in developing countries and small islands; in March 2022, they launched a five-year initiative to ensure the availability of early warning systems on a global scale (WMO, 2022). When predicting pluvial floods, high-resolution precipitation monitoring is crucial since an intense convective storm, whose diameter can be as small as a few kilometers, can evolve, precipitate, and trigger a flood within minutes (Peleg et al., 2022). Maintaining and deploying ground monitoring instruments, however, is a costly endeavor. To increase the accuracy of existing systems or extend their availability to cities where no system yet exists, one alternative is to qualitatively monitor the occurrence and structure of storms using low-cost acoustic rainfall sensors.

There are two questions that should be answered before considering deploying a low-cost acoustic rainfall sensor network and implementing it within an early warning system: how accurate are these sensors and what are their advantages/limitations with respect to other rainfall monitoring devices? In order to hint at the answers to these questions, we present laboratory and field results from three experiments at the rooftop, small-city and large-city scales. As a final point, we discuss the potential of using low-cost acoustic rainfall sensors for flood warnings.

## 2 Acoustic rainfall sensors

Acoustic rainfall sensors are simple devices. They usually consist of a small box (about 5-20 cm in size, Fig. 1b) with a flat-top surface (metal, plastic or skin) that acts like a drum when raindrops fall on it, a microphone, a power supply (usually a 1.5-3V battery), and a recording device (i.e., a logger). As acoustic sensors do not have mechanical components, they are less expensive to manufacture than, for example, tipping bucket rain gauges. It is possible to set up the audio recording device to record individual raindrops hitting the drum, or to record all the background noises continuously.

As early as the 1970s, acoustic sensors were used to monitor rainfall (Gray et al., 1974; Kampwerth and Rasmussen, 1974). Because the microphone quality was too low, resulting in low-accuracy measurements, and because the recording devices were too bulky and too expensive, acoustic sensors have not been widely used. But parts have become cheaper and smaller and open-source electronics platforms, such as Arduino, have made it possible to design and build small and low-cost acoustic rainfall sensors (Trono et al., 2012; Dunkerley, 2020). The development of acoustic methods for recording rainfall intensity





and occurrence is growing in popularity. For example, the sound recorded by opportunistic sensors in public spaces, such as security cameras, can be analyzed and converted to rainfall intensities (Dunkerley, 2022).

## 3 Monitoring rainfall using acoustic sensors

The purpose of this section is to illustrate how acoustic sensors can be used to monitor rainfall on a city scale (as evidenced in Benoit et al. (2018)'s pioneering study). We used Goodsell Systems' BIG-DRIP acoustic sensors (www.goodsellsystems.co.uk) for this purpose (Fig. 1a,b). The sensors are small (70 mm diameter by 28 mm high) and are able to record raindrop counts at intervals between 1 s and 24 h. Originally designed to monitor water drops in caves, the sensors have a maximum sensitivity of four drops per second. Besides the original sensors, we examined a newer version designed for outdoor use that can record rates higher than 20 drops per second.

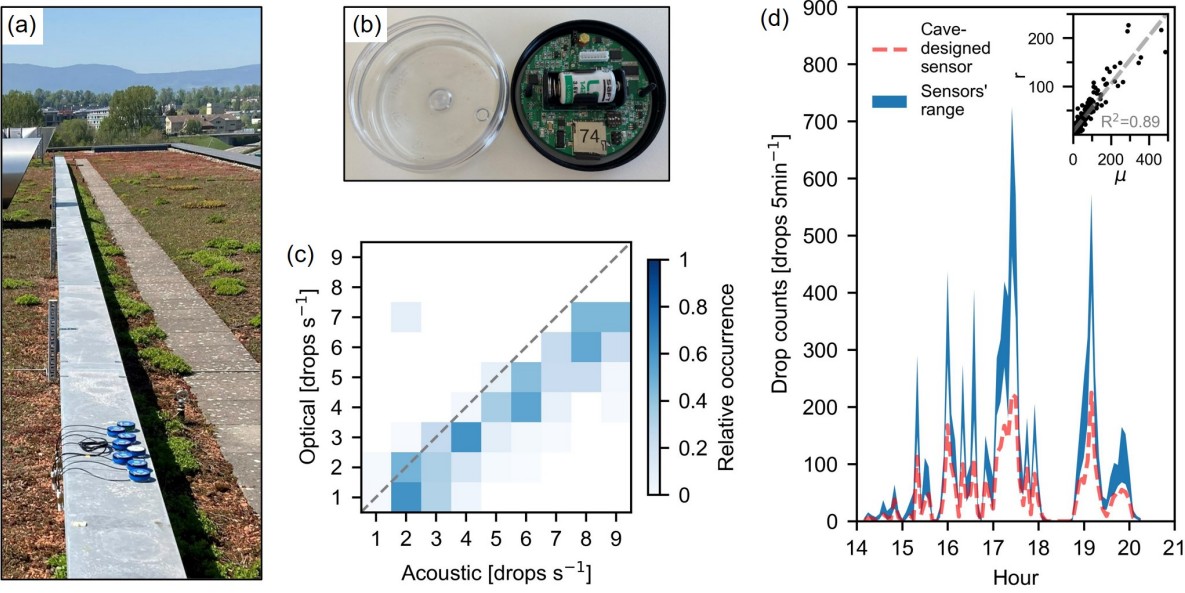

**Figure 1.** (a) The rooftop experiment with eight acoustic sensors placed closely together. (b) A close-up of an acoustic sensor. (c) Laboratory dripping experiment results; the occurrence of the acoustic sensors' records is in relation to the optical sensor's records. (d) Raindrops recorded by the seven high-sensitivity acoustic sensors on the roof of the University of Lausanne (shown in a) during a rainfall event on April 30[th], 2022 (blue area indicates maximum-minimum records). Red dashed line depicts the records from a single low-sensitivity (cave-designed) sensor. Inset shows the sensor range ($r$) in relation to mean drop counts ($\mu$) with a linear fit (gray line).

As a first step, we tested the sensors' ability to record drop rates against an independent digital counter in the laboratory under ideal conditions (e.g., no wind, full control of drip rate). Next, the recording uncertainty of the sensors was analyzed under natural conditions on the rooftop of the University of Lausanne by grouping several sensors together. Our last step was to deploy acoustic sensors in Zurich and Milan. In Zurich, seven sensors were deployed in various parts of the city, each





characterized by a unique Local Climate Zone (LCZ, as defined by Stewart and Oke, 2012). Each of the sensors was located next to Meteoblue's urban climate stations, which served as independent climate sensors (i.e., "ground truth" for validation). To estimate the sensors' capability to capture rainfall spatial variability at a large city scale, 30 climate sensors were deployed at 10 sites (2-3 sensors per site) in the city of Milan. No funnels were used to cover the acoustic sensors (e.g., Benoit et al.,
2018). The following subsections provide a brief summary of the results from the different sites and experiments, as well as a concise discussion of the sensors' capabilities and limitations.

### 3.1 Lab and rooftop experiments

Vernier's Drop Counter (*Go Direct Drop Counter*; GDX-DC), an optical device that counts drops passing its sensor over a given time interval, was used as a reference for the laboratory experiment. By controlling the dropping rates, the acoustic
sensors were tested for accuracy in recording drops between 1 and 9 per second. In the original version of the acoustic sensors (cave-designed, less sensitive microphones), recording results were unsatisfactory due to considerable underestimation of the number of drops (Fig. S1). By increasing the sensitivity of the microphone, the acoustic sensors were able to record the drops well (Fig. 1c); in fact, at high dripping rates (over 7 drops per second), the acoustic sensors outperformed the recording ability of the optical sensor.

Next, we assessed the sensors' uncertainty in recording raindrops under natural conditions. This was achieved by grouping the sensors together on top of the Géopolis building at the University of Lausanne, a few centimeters apart (Fig. 1a). We left the cave-designed sensors recording raindrops for three weeks, and then repeated the experiment with the high-sensitivity acoustic sensors, using a single cave-designed sensor as a proxy for comparison. We found that the raindrop record uncertainty linearly increases with rainfall intensity, thus the variability between maximum and minimum records can be meaningful in heavy
rainfall (Fig. 1d). As an example, the records for the most intense rainfall peak in Fig. 1d range between 500 and 700 drops per 5 min. For the cave-designed sensors, record uncertainty is lower (not shown), but this is because its measurement ability is limited by its lower record sensitivity (Fig. S1 and Fig. 1d).

The following implications can be drawn regarding the acoustic sensor's capability to monitor rainfall based on our laboratory and rooftop experiments: (i) individual drops falling at a high frequency can be recorded satisfactorily by the sensors
(Fig. 1c); (ii) the sensor's ability to record drops has been improved significantly by increasing the microphone sensitivity, and further improvements are possible in the future (Fig. S1 vs. 1c); (iii) with the current setup of the sensors, it would be advisable to have 2-3 sensors per location in order to assess and partly mitigate the recording uncertainty during intense rainfall (inset of Fig. 1d); (iv) it follows that the current sensor design prevents from easily converting drop rate into rainfall intensity. Nevertheless, it appears that the sensors can provide meaningful information about how rainfall magnitudes are distributed in space
and time; further, (v) the acoustic sensors capture the rainfall onset, cessation, and general temporal dynamics well (Fig. 1d).

### 3.2 Case studies: the cities of Zurich and Milan

The cities of Zurich and Milan were selected as case studies for testing the acoustic sensors at large spatial extents. Zurich is well monitored (a climate station per kilometer square) by Meteoblue, while Milan is a large city, relatively flat, and easily




accessible from the authors' hub. To have a sufficient number of sensors to monitor two case studies, we deployed both the

low-sensitivity (cave-designed) and the high-sensitivity acoustic sensors (in Zurich and Milan, respectively). We deployed the acoustic sensor networks in April-May 2022 (Fig. 2), choosing sites to represent the urban microclimate (i.e., representing different LCZ).

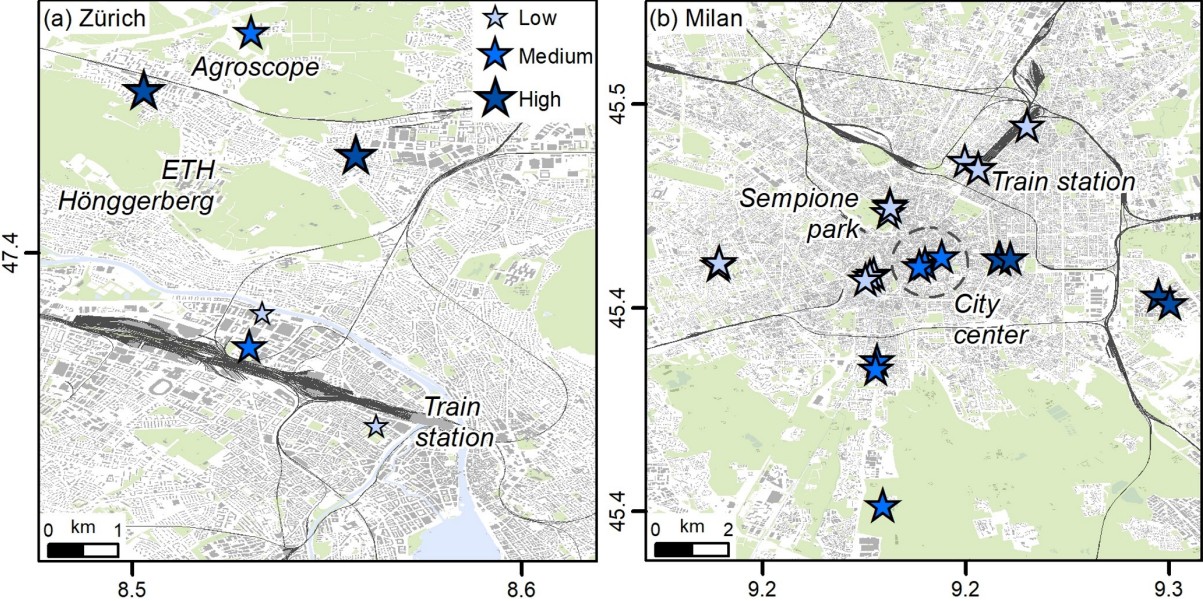

**Figure 2.** Locations of six low-sensitivity (cave-designed) acoustic sensors in Zurich (a; a single sensor is located at each site) and 29 high-sensitivity acoustic sensors in Milan (b; 2-3 sensors are located at each site). In Zurich, the size of the star indicates the Pearson correlation with nearby meteoblue AG stations (low correlation: 0.59-0.65, medium: 0.65-0.75, high: >0.75) and the color of the star indicates how many raindrops were counted in April 2022 (medium range: 28000-32000). In Milan, only the color of the star changes (no marker size difference) to indicates the number of raindrops in June 2022 (medium range: 8000-33000). Background city maps are based on © OpenStreetMap data, which is available under the Open Database License.

Upon installation, we assumed a 10-20% failure rate of the low-cost acoustic sensors due to hardware malfunctions. But the actual failure rate was much lower than expected, with only two low-cost sensors failing out of 37 deployed (5% failure rate)

for reasons unrelated to the power supply (e.g., microphone or logger failures).

By using Pearson correlation, we compared the temporal dynamics of the recorded raindrops by the acoustic sensors with the recorded rainfall intensities from the nearby meteoblue rain gauges in Zurich. Correlation values ranged from 0.59 to 0.77, which is satisfactory, especially since the results are for less-sensitive sensors. Interestingly, the lowest correlations were found in the vicinity of the train station, where fewer raindrops were recorded (Fig. 2a). In view of the limited number of sensors

deployed, and the correlation computed for only a single month (April 2022), this interpretation should be considered with caution.





Next, we demonstrate the ability of the acoustic sensors to capture rainfall spatial variability within cities. The analysis in Zurich focuses on the month of April, with four rainfall events (a maximum of 35000 raindrops counted in a single sensor), while in Milan, we examine the month of June, with five rainfall events (a maximum of 47000 raindrops counted in a single

sensor). In Zurich, two of the sensors indicate lower amounts of raindrops in densely built areas (along the train lines, Fig. 2a). In Milan, raindrops amounts are higher south- and east of the city center, with less rainfall falling over and west of the train station and Sempione park (Fig. 2b). It is clear from this example that even if acoustic sensors do not convert raindrops to rain intensity, they can still provide valuable information about where rain is falling and its general magnitude.

## 4   The added value of acoustic sensors in rainfall monitoring

In order to understand how acoustic sensors might contribute to rainfall monitoring, we must first examine alternative monitoring methods. Fig. 3a illustrates several monitoring devices that will be briefly discussed here (for a more comprehensive review see e.g., Uijlenhoet et al., 2018). Satellites monitor rainfall at large spatial ($10^1$ km) and temporal (few hours to days) scales, and indirectly estimate rainfall by sensing the cloud top temperatures or by evaluating absorption and scattering properties of the clouds. Weather radars remotely sense rainfall at much finer space-time scales ($10^0$ km and minutes); unlike satellites, they

rely on the backscattering of hydrometeors at atmospheric levels closer to the cloud base. The two methods can both be used to derive rainfall fields, which describe how rainfall evolves over time and space. Nevertheless, the spatial resolution of rainfall fields may be too coarse to represent small-scale variability in rainfall (Peleg et al., 2018) for hydrological processes relevant for urban flood forecasting (Ochoa-Rodriguez et al., 2015; Cristiano et al., 2017). In addition, rainfall estimates are not very precise since the rainfall is not sensed at ground level and the estimate is indirect. Rainfall intensity near the ground can be

estimated using commercial microwave links (CML) by quantifying the attenuation of the signal transmitted between cellular antennas. CML allow a high level of temporal accuracy (minutes), but the spatial resolution remains relatively coarse, often at the $10^1$ km scale. It is most accurate to estimate rainfall using rain gauges. Even though rain gauges can record rainfall intensity at a high temporal resolution of minutes, their sparse distribution makes it difficult to accurately represent rainfall intensity in space. Low cost acoustic sensors can be deployed with redundancy and in many places, in order to record raindrops where rain

gauges or commercial microwave links are not available, or where radar beams are blocked. Thanks to their simple design and small size they can be deployed easily and discretely on most flat-surfaces or objects, near the ground.

Can acoustic sensors provide useful information about rainfall? We found that acoustic sensors are capable of detecting well the onset and end of rainfall events. Furthermore, they can record raindrops at sub-minute intervals, allowing them to capture rainfall spatial variability at a higher temporal resolution than the other monitoring devices. There are, however, drawbacks

to their use, including the fact that they record the number of raindrops hitting the drum, and the conversion from drops per time interval to rainfall intensity (e.g., from drops per minute to mm per hour) is not straightforward. Our preliminary comparison with rain gauged data suggests that this conversion can be theoretically approximated, but further research is needed in this direction. A possible solution is to cover the acoustic sensors with funnels (e.g., Benoit et al., 2018), which will slow the dripping rate and increase the accuracy of the measurements and potentially also the conversion to rainfall intensity,




especially when examining rainfall rates at accumulation periods beyond the minute scale. Nevertheless, the high correlation we found between rainfall intensities (from rain gauges) and drop counts (acoustic sensors) implies that the differences in rainfall intensity between locations at a given time interval will be adequately reflected by the acoustic sensors. Another limitation is that the sensors cannot record solid precipitation, such as snow, but these are less important in triggering pluvial flooding. As another point to consider, the accuracy of the acoustic records can be affected by ground vibrations, for example if placed near

**Figure 3.** From rainfall monitoring to flood alerts. (a) Schematic illustration of the most common rainfall monitoring devices at different spatial and temporal scales. Real-time records from (a) are merged to form a representative rain field (b), which are then used in near-real-time nowcasting (c). The simulated nowcast rainfall from (c) is then fed into a hydrodynamic model to forecast urban floods in the short term (d), and if necessary, flood alerts are sent out (e).



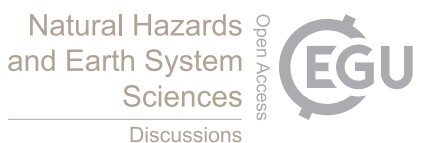

tram-stops, typically found in urban areas. However, this can also be solved by applying strict quality control to the data and removing white noise. Overall, it is our opinion that the advantages of using acoustic sensors to enrich an existing monitoring network or establish a new low-cost network outweigh these disadvantages.

## 5  Future implementation in early warning systems

The process of issuing a flood alert involves several steps (Fig. 3): (a) different monitoring devices are used in the area of
interest to observe rainfall; (b) by combining the information from the devices, a rainfall field is constructed that estimates how rainfall is spatially distributed at a given time; (c) a nowcasting model is applied to describe how rainfall evolves in space and time within a short time frame (e.g., an hour); (d) potential flooding is assessed using the nowcasted rainfall; and (e) public flood alerts are issued when necessary.

The rainfall merging process (Fig. 3b) can benefit from the use of acoustic sensors. Rainfall information, even if binary in
nature (i.e., yes/no rainfall), can contribute significantly to a better representation of rainfall's spatial structure. In addition, the merging process does not require the measurement units of the different monitoring devices to be unique. Therefore, even if rainfall information from acoustic sensors cannot be converted to rainfall intensity, it can still be merged. Acoustic sensors can also be deployed to support individual monitoring devices prior to merging (Fig. 3a). Low-cost acoustic sensors can, for example, be deployed on long commercial microwave links paths to improve the partitioning of rainfall intensities along the
line. Last, acoustic sensors can potentially be used directly in flood forecasting (Fig. 3c). Theoretically, using a sufficiently large number of acoustic sensors covering a large area within and around a city, as well as a sufficiently long training period, a machine learning model can be trained to forecast floods without the use of other rainfall monitoring devices. This could be a useful solution for many cities in low-income countries, for example, and could contribute to the WMO (2022) efforts to set up flood alerts worldwide. If acoustic sensors are to be used in future rainfall monitoring, further development of the
acoustic sensors will be necessary, including minimization of recording uncertainties and the ability to transmit data in real time. Following our examination into using simple low-cost acoustic sensors, we can see the potential they hold.

*Author contributions.* NP wrote the first draft of the manuscript and prepared the last figure. HTS deployed and maintained the sensor network, analyzed the data obtained from the sensors, and prepared the Figures 1 and 2. All authors equally contributed to revising and structuring the the manuscript.

*Data availability.* Data recorded by the acoustic sensors presented in this manuscript can be requested in writing from the corrensponding authors.



*Competing interests.* NP and FM are part of the editorial board of the special issue "Hydro-meteorological extremes and hazards: vulnerability, risk, impacts, and mitigation" in NHESS. The peer-review process was guided by an independent editor. The authors declare no other competing interests.

*Acknowledgements.* NP and HTS acknowledge the support of the Swiss National Science Foundation (SNSF), Grant 194649 ("Rainfall and floods in future cities"). FM was partially supported by CARIPARO Foundation through the Excellence Grant 2021 to the "Resilience" project. The authors thank meteoblue AG (www.meteoblue.com) for providing the rainfall data for the city of Zurich and Goodsell Systems Ltd for their assistance and further developing the used acoustic sensors. We also thank Marika Koukoula, Tabea Cache, Wenyue Zou, and Zhaozhao Zeng for their assistance deploying the sensors in Zurich and Milan.



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
