# Peer review of "Brief communication: the potential use of low-cost acoustic sensors to detect rainfall for short-term urban flood warnings"

_Natural Hazards and Earth System Sciences, 2022_

## Author Response (AR1)

February 1st, 2023

Prof. Paolo Tarolli
Editor
Natural Hazards and Earth System Sciences

RE: Paper NHESS-2022-257

Dear Prof. Paolo Tarolli,

We appreciate your handling of our manuscript. The revised manuscript entitled "*Brief communication: the potential use of low-cost acoustic sensors to detect rainfall for short-term urban flood warnings*" is enclosed, as well as supplemental materials and, as requested, a letter of response to all the remarks made by the reviewers. All the concerns raised by the reviewers have been addressed in the revised manuscript. We are confident that the manuscript can be considered for publication in *Natural Hazards and Earth System Sciences*.

Listed below are our responses (in blue) to the comments and suggestions of the reviewers. Line numbers refer to the "track changes" version of the manuscript.

Sincerely,

Nadav Peleg

On behalf of Herminia Torelló-Sentelles, Grégoire Mariéthoz, Lionel Benoit, João P. Leitão, and Francesco Marra

**Reviewer #1**

The authors present several experiments with low-cost acoustic sensors for urban rainfall monitoring and propose their use in short-term urban flood warning. The experiments consist of one lab and two outdoor experiments testing the accuracy and possible advantages and limitations of these sensors. Overall, these sensors do not yield exact rainfall amounts but rather can complement existing observations by giving begin and end of rainfall events as well as spatial rainfall distribution when a dense network is employed.

The topic of this brief communication suits well into NHESS' scope and is presented in a good quality. Despite the fact that acoustic sensors were introduced for rainfall measurements earlier as referenced by the authors, here these sensors are brought into perspective with other sensors and the real-world application of urban flood warning. I found some issues in the manuscript which should be addressed by the authors prior to publication which I recommend.

We would like to thank the referee for reviewing our paper and for her/his constructive comments and suggestions.

Specific comments

1. **Figure 2** - lack of time series. This are two issues which could potentially solved at once. First, Figure 2 is somewhat unintuitive because the same symbols are used to depict different things. Even with the figure caption being informative the Figure would benefit from a revision. Suggestions would be to remove the correlation and rain drop count to (a) new subplot(s) showing e.g. the correlation to rain gauges over distance. Also, the rain gauges from meteoblue could be shown in the map. Second, I'd really like to see time series of acoustic sensors compared to reference rain gauges. Such time series could illustrate both the correlation presented in Fig. 2 and the text as well as some issues why these sensors cannot be used to derived rainfall amounts directly. Time series plots could be shown as supplementary material, while I would encourage the authors to add them to Figure 2.

The reviewer has provided great suggestions, which we completely agree with. To begin, we separate the correlation analysis (now shown in Fig. S2) from the original Fig. 2. In Fig. 2c, we now present an example of a single rainfall event with good agreement in both timing and magnitude between acoustic sensors and the nearby meteoblue AG rain gauge (distances between the devices are now reported in Table S1). Further, in Fig. S3, we added two additional rainfall events, which demonstrate that while the timing is consistent between the devices, the acoustic sensors tend to overestimate the peak rainfall. Last, we included an example of a rainfall event recorded in Milan (Fig. 2d) to illustrate the potential of acoustic sensors for capturing the space-time variability of rainfall.

In addition to editing the figure caption in Fig. 2, we added the following paragraph in Section 3.2: "In addition, we compared the rainfall timing and magnitude between the raindrops recorded by the acoustic sensors and the rainfall depth recorded by the nearby meteoblue AG rain gauges (distances are reported in Table S1). In regard to the timing of rainfall events, we found that both rain gauges and acoustic sensors recorded the onset and cessation of rain concurrently (Fig. 2c and Fig. S3). We found a good qualitative agreement between raindrops recorded by the acoustic sensors and rainfall intensities recorded by the gauges (e.g., Fig. 2c). Occasionally, the acoustic sensors overestimate the peak rainfall during an event (Fig. S3). By analyzing the timing and magnitude of rainfall recorded in multiple locations by the acoustic sensors in Milan (see example in Fig. Fig. 2d), it is apparent that the sensors are capable of capturing the space-time variability of rainfall. It is clear from these examples that even if acoustic sensors do not convert raindrops to rain intensity, they can still provide valuable information about where rain is falling and its general magnitude".

2. **More specific information on deployment.** While the whole process chain from the acoustic sensor to a warning system is depicted in the manuscript some description on the acoustic sensors regarding pricing compared to traditional sensors, setup and the setup in a real-time, operational way with many sensors (as envisioned in l. 165)

would make this manuscript more inspirational for researchers and stakeholders playing with the thought of experimenting or deploying such systems.

We agree with the reviewer and have added the following information to the text. Line 55: "As a rough comparison, the price of a low-cost acoustic sensor today is less than 100 USD, whereas the cost of a commercial tipping bucket exceeds 500 USD". Line 153: "Furthermore, acoustic sensors are relatively easy to deploy and maintain since they do not need balancing or frequent cleaning, as are required for example when deploying rain gauges".

3. **Questions raised in the manuscript.** Two questions are raised at the end of the introduction and answered in the following chapters while a third one which is a mix of both is raised in l. 137. The easiest way to make this more consistent would be to add this third question to the intro, but there might be other ways to solve this inconsistency.

We have revised the text at the end of the introduction to include the following three questions: "There are three questions that should be answered … how accurate are these sensors, what are their advantages/limitations with respect to other rainfall monitoring devices, and ultimately, what value can acoustic sensors add to rainfall monitoring?".

Technical issues
4. Line 24. You could also list examples of ground observations.
We added "rain gauges" as an example of ground monitoring devices.

5. Line 68. I assume you mean 30 acoustic sensors?
Thank you for noticing this. The text has been corrected accordingly.

6. Line 131. E-band CMLs are also in the magnitude of $10^0$ km and can deliver data with sub-minute resolution.
We added the following to the text in response to the reviewer's comment: "...exceptions are E-band CML, which are in 1 km scale".

Further issues
7. The article type allows for 20 references while the authors cite 22 references. In my opinion 22 would be fine.
There are now 23 references (an additional reference was added during the revisions). We believe all should be maintained; in accordance with the reviewer's opinion.

8. Data availability is not in agreement with NHESS data policy, e.g. "If the data are not publicly accessible, a detailed explanation of why this is the case is required." The FAIR way of course would be a publication of acoustic sensors data accompanied by reference rainfall data but I can understand if the latter one is not possible due to meteoblue's data policy.
In the "Data availability" section, we now link readers to the Zenodo file repository that hosts the rainfall data from both the acoustic sensors and meteoblue AG's rain gauges.

**Reviewer #2**

The manuscript presents performance of a drop counting acoustic rain gauge which could according to authors in future contribute to the improvement of short-term flood warning systems. The authors are addressing two questions. First, how accurate are BIG-DRIP acoustic sensors, and second, what are their advantages/limitations in comparison to other rainfall monitoring devices. The paper presents first results from a an extensive evaluation study of low-cost drop-counting acoustic sensors. The presented results are, unfortunately, insufficient to reach substantial conclusions, especially when it comes to the paper main topic - the evaluation of these sensors to flood early warning. My major criticism concerns both the contents of the paper and the level of detail of the evaluation analysis. Furthermore, the description of data used is not sufficient to allow reproduction and interpretation of the results. Finally, the advantages/limitations of the sensor are discussed only in very general way without stronger link to the presented results and without context with respect to already established acoustic sensors.

We appreciate the reviewer's time and effort as well as their constructive and positive feedback. In the reviewer's opinion, the study's evaluation analysis and data description are lacking in detail, and the discussion of the advantages and limitations of using acoustic sensors in flood early warning systems remains rather general. We completely agree with the reviewer on this evaluation, but we have to point out that this article is submitted as a "brief communication". Our goal here is not to discuss the capabilities of a specific type of acoustic sensor (i.e., the BIG-DRIP), but rather to highlight to the natural hazard community the potential of using acoustic sensors in general to aid existing or planned urban flood warning systems. We complement our commentary with a short quantitative demonstration of the capabilities and current limitations of one type of acoustic sensor.

In response to the referee's comments, we revised our manuscript in accordance with his/her suggestions while preserving the intended commentary angle and keeping the manuscript concise and within a "brief communication" format. Several modifications have been made to the manuscript, including:

1. By adding new plots to Fig. 2 and adding two additional figures to the supplementary information, we demonstrate how well the acoustic sensors can record rainfall timing and magnitude and their ability to represent the rainfall space-time variability.
2. A brief discussion was added on possible approaches for converting drop counts to rainfall intensity. That includes the effect of the drop size distribution and the possibility of segmenting rainfall events into rain types.
3. To ensure reproducibility and allow comparison with other studies, the data collected by the acoustic sensors have now been published in an open repository (Zenodo).
4. As suggested by the reviewer, we extended the discussion in the text on the advantages and limitations of using acoustic sensors, in particular the possibility of using these sensors for a longer period (maintenance and energy consumption).

Specific comments

1. **To the paper contents:** the title and abstract of the paper do not unambiguously reflect contents of the paper.
The title has been revised to better reflect the content of the manuscript: "Brief communication: the potential use of low-cost acoustic sensors **to detect rainfall for** short-term urban flood warnings". Furthermore, we have slightly revised the abstract: "…A fast and reliable **rainfall detection system** should thus be implemented in flood-prone cities to warn the public of upcoming floods and save lives and reduce damage…".

2. The presented results are insufficient for evaluating the sensors' potential for flood early warning. The authors discuss this potential in a separate section; nonetheless, the discussion is very general and does not build on presented results. The authors conclude that i) acoustic sensors can significantly contribute to representation of rainfall spatial structure with binary (yes/no rainfall) information, ii) that they can support other devices, e.g. support reconstruction of spatial distribution of rain rate along microwave links, and iii) that they can be used directly in flood forecasting using some (not specified) machine learning technique. However, none of these topics (although being

relevant) is specifically investigated in the study. The presented results showing correlations between acoustic-rain-gauge observations (drop counts/time) and near-by standard rain-gauge observations (rainfall intensities) cannot be considered an evaluation of rainfall spatial structure. The authors conclude they see the potential in the use of low-cost acoustic sensors based on their study. Such conclusion is unfortunately not supported by presented results.

We agree with the reviewer that the evaluation of the acoustic sensors in the original manuscript was not sufficiently broad. We, therefore, extended the comparison between the raindrops measured by the acoustic sensors and the rainfall intensity measured by the nearby meteoblue AG rain gauges. In Fig. 2c, we now present an example of a single rainfall event with good agreement in both timing and magnitude between the two devices. Further, in Fig. S3, we added two additional rainfall events, which demonstrate that while the timing is consistent between the devices, the acoustic sensors tend to overestimate the peak rainfall. Last, we included an example of a rainfall event recorded in Milan (Fig. 2d) to illustrate the potential of acoustic sensors for representing the space-time variability of rainfall. We also added the following text in Section 3.2: "In addition, we compared the rainfall timing and magnitude between the raindrops recorded by the acoustic sensors and the rainfall depth recorded by the nearby meteoblue AG rain gauges (distances are reported in Table S1). In regard to the timing of rainfall events, we found that both rain gauges and acoustic sensors recorded the onset and cessation of rain concurrently (Fig. 2c and Fig. S3). We found a good qualitative agreement between raindrops recorded by the acoustic sensors and rainfall intensities recorded by the gauges (e.g., Fig. 2c). Occasionally, the acoustic sensors overestimate the peak rainfall during an event (Fig. S3). By analyzing the timing and magnitude of rainfall recorded in multiple locations by the acoustic sensors in Milan (see example in Fig. Fig. 2d), it is apparent that the sensors are capable of capturing the space-time variability of rainfall. It is clear from these examples that even if acoustic sensors do not convert raindrops to rain intensity, they can still provide valuable information about where rain is falling and its general magnitude".

Regarding the reviewer's points (ii) and (iii), it is important to note that we intend only to discuss the potential (future) use of acoustic sensors for improving rainfall estimates derived from CML, as well as their potential application in flood forecasting by means of machine learning. The reviewer noted that these topics (although relevant) are not specifically addressed and in-depth studied in the paper. The purpose of mentioning them is to encourage the reader to think of potential future applications for acoustic sensors, but further investigation of these possibilities is beyond the scope of this short commentary.

The referee commented that "the presented results showing correlations … cannot be considered an evaluation of rainfall spatial structure". This statement is true, and we have therefore added the timing and magnitude analysis mentioned above. Additionally, we realized that we used the term "rainfall spatial structure" rather than "rainfall spatial variability" in line 187, and this has been corrected. Furthermore, the new Fig. 2d illustrates how acoustic sensors can record rainfall spatial and temporal variability using a single rainfall event as an example.

Considering the additions we have presented, we believe it is reasonable to conclude that low-cost acoustic sensors should be further investigated for applications that require monitoring rainfall space-time variability.

3. **To the evaluation analysis:** The analysis evaluating accuracy of drop-counting sensors presents only very preliminary results showing i) correlation coefficients between detected rain drop counts and rain rates observed during one event at roof top of Authors' hub, and ii) again correlation coefficients between detected raindrop counts and rain rates observed by near-by rain gauges during two case studies. Surprisingly the authors do not discuss at all how drop counts can be converted to rainfall intensity and how this is affected by drop sizes. The effect of drop size distribution is not discussed at all. The presented analysis is simply insufficient to soundly address research questions formulated by the authors in the introduction. How accurate is the transformation between drop counts and rain rate in terms of systematic deviation, or error mean square error? How does the accuracy relate to drop size distribution? What are detection limits of drop sizes, how well the sensors perform during heavy rainfall? How accurate is binary information (is rain / no rain)? How well can the sensor detect onset of an event (this is mentioned but not presented)? These (and other) questions would help to formulate more specific conclusion about accuracy of the sensors.

According to the referee: "… the authors do not discuss at all how drop counts can be converted to rainfall intensity …". This is actually discussed in Section 4. As the reviewer pointed out, we did not discuss the effect of drop size distribution in the original manuscript. This issue has now been introduced in the text in line 161: "In particular, the effect of drop size distribution on this conversion should be explored. There is also a need to investigate the relationship between the type of rainfall (e.g., convective or stratiform) and the drop-intensity conversion. The acoustic sensors can also be covered with funnels…". We agree with the reviewer that we do not provide an exact answer here regarding the accuracy of the raindrops' measurement by the acoustic sensors. Nevertheless, the purpose of this manuscript is to raise awareness of the potential use of acoustic sensors in rainfall monitoring and their incorporation into future applications. The reviewer's questions are all pertinent and should be addressed in future research.

4. **To the description of the data and evaluation methods:** The level of detail provided about material and methods is insufficient to ensure reproducibility and enable comparison with other studies. For example, it is not clear how far are the reference rain gauges from drop counting sensors during case studies, what type of rain gauges these are, at which temporal resolution is the data evaluated. There is also no information about events occurring during the evaluated period, etc.

In the "Data availability" section, we now link readers to the Zenodo file repository that hosts the rainfall data from both the acoustic sensors and meteoblue AG's rain gauges. We also added the information regarding the distances between the acoustic sensors and the closest meteoblue AG rain gauges in Table S1. Last, we added the following information regarding the rain gauges in line 71: "Each of the sensors was located next to meteoblue AG's urban climate stations (equipped with tipping bucket rain gauges - 0.2 mm per tip, and recording rainfall depths at 15 min intervals) …".

5. **Advantages and limitations** of the sensors are discussed in section 4, however, in very general manner. The reader can learn that authors identified relatively strong correlation link between recorded drop counts and rain rates and that the sensor can detect onset of events. The second claim is not supported by provided results (no such evaluation presented). Sensors characteristics related to its operation in longer term, such as energy consumption, are not discussed. Sensor's performance with respect to already used acoustic sensors (e.g. https://www.vaisala.com/sites/default/files/documents/RAINCAP_Technology.pdf) might be also worth to discuss.

As mentioned above, we have added in the manuscript a reference to the sensors' ability to detect the onset of rainfall events. Additionally, we added the following in line 171 to address the sensors' operational issues following the suggestions of the reviewer: "The acoustic sensors require constant maintenance when it is deployed for a prolonged period of time, just like other monitoring devices. As an example, the acoustic sensors we used in this study require batteries to be replaced every two months (if measuring drop counts at 1 min intervals). A checkup and regular maintenance of the deployed sensors is recommended every month or so, but this is no different from what would be expected for other monitoring devices, such as ground climate stations. Moreover, while acoustic sensors are relatively easy to place, they can also be deployed in cities only during periods when intense short-duration rainfall bursts are anticipated, thus reducing some of the maintenance needs". In response to the reviewer's suggestion to compare BIG-DRIP's performance to the performance of other acoustic sensors, we found that this was outside the scope of this manuscript. Our objective is not to highlight the use of a specific sensor (BIG-DRIP in this case) but to demonstrate the potential for using low-cost sensors in general for rainfall detection applications.

In a view of my criticism, I cannot recommend the manuscript for a publication. I am convinced that more extensive analysis and better evaluation framework is required to reach more specific and scientifically relevant conclusions. Also the scope of the manuscript has to be better defined. The assessment of sensor's accuracy is a legitimate scope, however, it is not the evaluation of the sensor's potential for flood early warning.

Once again, we would like to thank the reviewer for his/her time and efforts. In their second reading, we hope the reviewer will be convinced of the manuscript's relevance to the NHESS readers based on the revised text and our clarification of the manuscript's aim.

Again, we would like to thank the Editor, Prof. Paolo Tarolli, and the two anonymous reviewers for their comments and suggestions.

Sincerely,

Nadav Peleg

On behalf of Herminia Torelló-Sentelles, Grégoire Mariéthoz, Lionel Benoit, João P. Leitão, and Francesco Marra